# Fluid–Structure Interaction Analysis in Ball Bearings Subjected to Hydrodynamic and Mixed Lubrication

Marvelúcia Almeida [1,†] , Flávia Bastos [1,*,†] and Sara Vecchio [2,†]

1   Graduate Program in Computational Modeling, Federal University of Juiz de Fora,
    Juiz de Fora 36036-900, Brazil; marvelucia.almeida@estudante.ufjf.br
2   Department of Mechanical Engineering, Federal Institute of Science Education and Technology of
    Southeastern Minas Gerais, Juiz de Fora 36080-001, Brazil; sara.vecchio@ifsudestemg.edu.br
*   Correspondence: flavia.bastos@ufjf.edu.br
†   These authors contributed equally to this work.

**Abstract:** The mathematical and computational modeling of the lubricated contact between bearing surfaces is presented to analyze the sliding friction using a realistic 3D model on a microscopic scale. The fluid–structure interaction model evaluates the effects of lubricant film thickness on friction in hydrodynamic and mixed lubrication regimes. Higher contact pressures are seen at the peaks of asperities, especially during mixed lubrication, in which the fluid volume is smaller. Calculated friction coefficients from a homogenization procedure of shear and normal forces in the hydrodynamic and mixed lubrication, close to 0.0045 and 0.014, respectively, were accurate and within the range specified in the Stribeck curve. Results demonstrate the computational model allows examining the effects of lubrication on contact between rough surfaces.

**Keywords:** computational modeling; finite element method (FEM); rough surface contact; lubrication; friction analysis

## 1. Introduction

Bearings are machine components that support rotating machine elements, especially axles, and wheels. Elements in rotating contact such as balls, cylindrical, or taper rollers, for example, transmit the main load instead of the sliding contact present in other types of sliding bearings [1]. Its application is extensive in industrial machinery of the most varied segments, moreover, speed multipliers operating in hydraulic and wind turbines, and speed reduction boxes located in automobile engines.

Bearings are responsible for 76% of premature problems in turbine gearboxes [2,3]. Generally, turbine runtime is expected by design for about 20 to 30 years. However, mostly 20% of the turbine's nominal life-bearing failures are recorded. The defects are related to axial shear failure due to heavy and dynamic loads, vibrations, lack of lubrication, and sudden temperature changes [4,5].

The failure of the main bearing in the motor shaft causes significant production loss in wind turbines, automotive and electrical equipment, and auxiliary devices [2]. In offshore wind turbines designed for a life of more than 20 years, operation and maintenance costs account for 20% to 25% of total revenues. Also, the failure of one element can generate losses in others. So, in effect, a USD 5000 bearing failure in a wind turbine can cost a USD 250,000 project, which will involve cranes, a service crew, gearbox replacements, generator rewinding, and downtime and loss of power generation, which also represent increased cost loss [6–8].

Experimental studies have been developed to investigate the causes of wind turbine gearbox bearing failures evaluating inclusions found at damaged sites by tomography [9–12].

Other studies [13–22] focused on the causes of bearing failures through analytical, numerical, and experimental methodologies, which analyze in more detail the wear and its

causes in the rolling motion, taking into account the surface roughness and the presence of lubrication on the parts in contact. According to Burstein [13] lubrication effectively reduces friction and heat losses and extends the life of the parts in contact.

Khonsari et al. [14] and Blau [15] the importance of evaluating the interaction effects between friction, lubrication, wear, roughness, plastic deformation, contact pressure, oxidation, changes in geometry at contact, and crystallographic reorientation. Zapletal et al. [16] pointed out the connection between lubricating film thickness and friction of a uniform surface texture along with the transition from hydrodynamic (EHD) to mixed lubrication in non-compliant contacts. Reichert et al. [17] investigated the impact of surface flattening on the friction coefficient using the mixed lubrication model at the microscale with the finite element software Abaqus. Larsson [18] presented a model of lubrication in all regimes applied to thrust bearings and wet clutches for analyzing surface roughness influence in contact mechanics when surfaces are boundary lubricated, as well as hydrodynamic film formation.

LorPusterhofer and Grün [19] evaluated the durability behavior of lubricated sliding contacts using a combined experimental (Ring-on-Disc) and computational model (journal-bearing geometry) to predict the tribological system destabilization. Guegan et al. [20] investigated the influence of smooth and rough surfaces on friction in an elastohydrodynamic contact (transition of EHD to mixed lubricated), using a ball-on-disc tribometer. They measured a range of slide–roll ratios (SRR) at different speeds with two oils of quite different viscosities, enabling experiments to be conducted over a wide range of lambda values, defined as the ratio between the film thickness and root-mean-square roughness of the surfaces (RMS). Farfán-Cabrera and Gallardo-Hernández [21] analyzed wear through a variation of the microscale abrasion test adopting the elastohydrodynamic lubrication regime (EHL) and elliptical contact geometric characteristics. Niu and Zhang et al. [22] examined the tribo performance in the starved lubrication condition on a non-conforming contact between laser-textured medium carbon steel surfaces.

There are many works in the literature that explore the rheological behavior of lubricating oil and grease. Using a coupled simulation model of fluid–structure interaction (FSI) based on the computational fluid dynamics method (CFD), Liu et al. [23] analyzed the lubricating characteristics of oil flow in bearings according to the increase of lubricating oil viscosity and its velocity, the angle and number of the fluid inlet nozzle. Morales-Espejel et al. [24] investigated the grease-lubricated region of bearings operating at low speed to devise a method that better considers the lubrication quality parameter according to the effects of the grease thickener and not only on the viscosity of the base oil. Laurentis et al. [25] analyzed the connection between various grease components (thickener and base oil) applied in bearings with the sliding friction generated in lubricated contacts.

Hartinger et al. [26] introduced a CFD approach for solving EHL lubrication in metal-on-metal contact using the freely available package OpenFOAM. They considered the isothermal and thermal cases under moderate loads and steady-state conditions, the phenomenon of cavitation under constant velocity, different viscosities, and slide-to-roll ratios. Vladescu et al. [27] studied the friction properties of a range of modifier-containing oils in an engine bearing under a hydrodynamic regime using a combined experimental and modeling approach (a journal bearing machine (JBM) supplied by PCS Instruments and a finite difference-based 3D thermal hydrodynamic lubrication model). Raisin et al. [28] investigated the influence on the tribological parameters of lubricated contact parts due to the presence of diamond-like carbon coatings. Their finite element numerical model considered variables such as drag velocity and infinite sliding conditions, generating temperature and thermal viscosity profiles for analysis and discussion about film thickness and friction. Vergne [29] evaluated the emergence of super-low frictional forces that occur in fluid–structure interactions when applying representative normal velocities and loads by simulating lubricated contacts as they exist in real life, which involved engineered surfaces and materials, EHL and EHD lubrication, and Stribeck diagrams.

Several experimental and numerical studies using the Finite Element Method (FEM) have been performed to analyze fatigue damage in rolling contacts. Vrbka et al. [30] verified the influence of microcavities on the surface texture and surface fatigue on a non-conforming contact under a mixed lubrication regime. Yan et al. [31] and Lorenz et al. [32] analyzed how roughness parameters such as mean roughness, mean square roughness, skewness, and kurtosis interfere with the fatigue life of rolling contacts under a mixed EHL lubrication regime. Lorenz et al. used a continuous damage mechanics (CDM) FEM model to estimate the fatigue life of machine elements. Shi et al. [33], using a non-Gaussian surface simulation technique and FEM stress analysis, calculated the relative fatigue life of bearing with a particular roughness profile subjected to high speed and loads. Finally, Toumi et al. [34], through numerical simulation of a three-dimensional dynamic and cyclic loading model and experimental comparison, evaluated the damage caused by contact fatigue of bearings.

This paper aims to present a 3D microscopic scale computational model implemented in a finite element program (Abaqus®) and to analyze, through fluid–structure Interaction (FSI) method, stresses, pressures, and friction between the bearing parts in contact. The work by Reichert et al. [17] is closer to this one in terms of methodology, but we can highlight as a main difference the fact that they have dedicated mainly to mixed lubrication conditions while this paper seeks to elucidate the differences between the two regimes: hydrodynamic and mixed.

This study focuses on a more theoretical and straightforward view of tribological systems, considering two widely studied lubrication regimes. The pervasive academic background of modeling friction phenomena and practical achievements served as a basis and motivation to initiate studies in the area.

Table 1 summarizes the findings of the main authors related to this work. It consists of data that were used in some way in the work, both in the adjustment of the model and for comparison and discussion of the results.

**Table 1.** Summary of findings of the main authors related to this work.

| Authors | Lubricating Regime | Lubricant Type | Viscosity (N.s/mm$^2$) | Solid Material | Roughness | Sliding Speed (mm/s) | Friction Coefficient |
|---|---|---|---|---|---|---|---|
| [13] | hydrodynamic contact | SAE 30–50 | 3 and 0.3 (at 100 °C) | - | 1.5 of lower surface and 3 of upper surface | 8900 | 0.5 to 0.8 |
| [14] | dry and lubricated contact | oil SAE 30 | - | - | - | 100 and 200 | 0.5 to 0.8 |
| [17] | mixed lubrication | - | 0.00313 (at 24 °C) | Ti–6Al–4V, AISI 1045 and 42CrMo4 | 0.704 μm. | 1000 | Global 0.042, Solid–solid 0.036 Solid–fluid part 0.006 |
| [19] | hydrodynamic micro model | Engine oil | 0.0055 (at 100 °C) | AlSn6 steel | - | 300 to 500 | 0.001 and 0.0005 |

**Table 1.** *Cont.*

| Authors | Lubricating Regime | Lubricant Type | Viscosity (N.s/mm$^2$) | Solid Material | Roughness | Sliding Speed (mm/s) | Friction Coefficient |
|---|---|---|---|---|---|---|---|
| [20] | elastohydrodynamic contact (transition of EHD to mixed lubricated) | oil | 0.2337 (at 24 °C) and 0.0243 (at 100°C) | steel | - | 20 to 2000 | 0.01 |
| [21] | elastohydrodynamic lubrication (EHL), boundary | engine oil Quaker States SAE 15W-40 | 0.1204 (at 24 °C) and 0.01571 (at 100 °C) | Cu–Al, Pb–Cu–Al, Sn–Al–Si | - | 147 and 10,000 | - |
| [22] | mixed lubrication and dry friction | PAO4 oil | 0.168 (at 24 °C) and 0.0039 (at 100 °C) | carbon steel | $R_a = 0.02$ µm | 100 | 0.11 to 0.6 |
| [24] | elastohydrodynamic lubrication (EHL) | Seven types of grease | - | steel | $R_q < 10$ nm | 6.3 | - |

## 2. Mathematical Formulation

The study is based on contact mechanics for linear elastic solids and on the Lagrangian-Eulerian coupled method for incompressible Newtonian viscous fluid in lubrication regimes.

### 2.1. Continuum Mechanics

The mathematical formulation that describes physical problems is the basis of computational numerical models. In this study, the continuum mechanics regarding a solid volume and a fluid describes the problem addressed. The used description comes predominantly from the literature of [35].

It is assumed a continuous medium consisting of a linear elastic solid material, subject to mass forces $\mathbf{b}(\mathbf{x}, t)$ inside its volume $V$ and surface forces $\mathbf{t}(\mathbf{x}, t)$ on its boundary $\partial V$. Thus, the governing equations to evaluate displacements $\mathbf{u}(\mathbf{x}, t)$, strains $\boldsymbol{\varepsilon}(\mathbf{x}, t)$ and stresses $\boldsymbol{\sigma}(\mathbf{x}, t)$ through time $t$ are:

$$\left. \begin{array}{r} \boldsymbol{\nabla} . \boldsymbol{\sigma}(\mathbf{x}, t) + \rho_0 \mathbf{b}(\mathbf{x}, t) = \rho_0 \dfrac{\partial^2 \mathbf{u}(\mathbf{x}, t)}{\partial t^2} \\ \boldsymbol{\sigma}(\mathbf{x}, t) = \lambda Tr(\boldsymbol{\varepsilon})\mathbf{1} + 2\mu\boldsymbol{\varepsilon} \\ \boldsymbol{\varepsilon}(\mathbf{x}, t) = \boldsymbol{\nabla}^S \mathbf{u}(\mathbf{x}, t) = \dfrac{1}{2}(\mathbf{u} \otimes \boldsymbol{\nabla} + \boldsymbol{\nabla} \otimes \mathbf{u}) \end{array} \right\} \begin{array}{l} \text{Cauchy's Equation} \\ \text{Constitutive Equation} \\ \text{Geometric Equation} \end{array} \quad (1)$$

The solid boundary $\Gamma \equiv \partial V$ considers the displacement boundary $\Gamma_u$ and the stress boundary $\Gamma_\sigma$ as the following:

$$\left. \begin{array}{l} \Gamma_u : \mathbf{u} = \mathbf{u}^* \\ \Gamma_\sigma : \mathbf{t}^* = \boldsymbol{\sigma} \cdot \mathbf{n} \end{array} \right\} \text{Boundary Conditions in Space} \quad (2)$$

The initial and boundary conditions are:

$$\left. \begin{array}{rcl} \mathbf{u}(\mathbf{x}, 0) & = & \mathbf{0} \\ \dot{\mathbf{u}}(\mathbf{x}, 0) & = & \mathbf{v}_0 \end{array} \right\} \text{Initial Conditions} \quad (3)$$

$Tr(\boldsymbol{\varepsilon})$ corresponds to the trace of the strain tensor $\boldsymbol{\varepsilon}$, $\lambda$ and $\mu$ are known as Lamé constants, which characterize the elastic behavior of the material and are obtained experimentally. In addition, $\mathbf{1}$ designates a unit tensor of a second-order tensor. Finally, $\boldsymbol{\nabla}$ is the gradient or Nabla operator, and $\rho_0$ is the density or specific mass of the linear elastic solid. The operator $\otimes$ denotes a tensor product since $\mathbf{u}$ is a vector, also referred to as a first-order tensor. The fixed boundary condition is also referred to as a Dirichlet condition, which

would be the prescribed displacement condition, while the state described by the derivative of the displacement is also known as a Neumann condition.

We assume a continuous medium consisting of an incompressible Newtonian viscous fluid in the barotropic and transient regime, subject to viscous and inertial forces within its material volume $V_t$ and surface traction or pressure forces at its boundary $\partial V_t$.

The fluid is a particular case of a continuous medium, described by its constitutive equations. Thus, the equations that analyze density $\rho_f(\mathbf{x}, t)$, velocity $\mathbf{v}_f(\mathbf{x}, t)$, stresses $\sigma_f(\mathbf{x}, t)$, and pressures $p_f(\mathbf{x}, t)$ through initial conditions $t = 0$ and time $t$ outline the fluid mechanics problem.

The boundary conditions of the fluid mechanics problem are related to the spatial or Eulerian description when analyzing a fixed control volume in space. There is the boundary condition with the velocity value determined in some boundary parts $\Gamma_v$ and the adhesion condition due to the viscous nature of the fluid. Furthermore, there is an impenetrability condition in some parts of the boundary (walls) $\Gamma_{v_n}$. In addition to these conditions concerning the fluid velocity, there are boundary conditions $\Gamma_\sigma$ associated with stresses or pressures. According to the problem, the thermodynamic pressure or the pressure in the fluid can describe the stress vector. Thus, it will represent a portion of the normal component of the stress vector $\mathbf{t}_f$ over a section of the boundary $\Gamma_p$.

Therefore, the equations that best govern the fluid mechanics problem, taking into account an incompressible Newtonian viscous fluid in the barotropic and transient regimes, are:

$$
\left.
\begin{aligned}
\boldsymbol{\nabla} \cdot \mathbf{v}_f &= 0 \\
-\boldsymbol{\nabla} p_f + \mu_f \boldsymbol{\Delta} \mathbf{v}_f + \rho_f \mathbf{b}_f &= \rho_f \frac{d\mathbf{v}_f}{dt} \\
\sigma_f &= -p_f \mathbf{1} + 2\mu_f \mathbf{d} \\
\rho_f &= \rho_f\left(p_f\right)
\end{aligned}
\right\}
\begin{aligned}
&\text{Continuity Equation} \\
&\text{Navier–Stokes Equation} \\
&\text{Constitutive Equation} \\
&\text{Kinetics of State Equation}
\end{aligned}
\tag{4}
$$

$$
\left.
\begin{aligned}
\mathbf{v}_f(\mathbf{x}, t) &= \bar{\mathbf{v}}(\mathbf{x}, t) \quad \forall \mathbf{x} \, \epsilon \, \Gamma_v \\
\mathbf{v}_n(\mathbf{x}, t) = \mathbf{v}_r \cdot \mathbf{n} &= \left(\mathbf{v}_f - \mathbf{v}^*\right) \cdot \mathbf{n} = 0 \quad \forall \mathbf{x} \, \epsilon \, \Gamma_{v_n} \\
\mathbf{v}_r(\mathbf{x}, t) = \mathbf{v}_f - \mathbf{v}^* = 0 &\Rightarrow \mathbf{v}_f = \mathbf{v}^* \quad \forall \mathbf{x} \, \epsilon \, \Gamma_v \\
p(\mathbf{x}, t) &= p^*(\mathbf{x}, t) \quad \forall \mathbf{x} \, \epsilon \, \Gamma_p
\end{aligned}
\right\}
\text{Boundary Conditions}
\tag{5}
$$

where $\mathbf{b}_f$ represents the vector of mass forces acting on the fluid, and $\mu_f$ is the constant of proportionality. It can be called the first viscosity coefficient, absolute viscosity, or dynamic viscosity. $\bar{\mathbf{v}}(\mathbf{x}, t)$ corresponds to the prescribed value of the velocity at a given point on the boundary, representing the Dirichlet boundary condition. $\mathbf{v}_n$ represents the component of the fluid rate in the direction normal to the border. $\mathbf{v}_r$ is the relative fluid-wall velocity, and $\mathbf{v}^*$ is the impermeable-wall velocity.

The general solution of the equations consists primarily in determining the hydrodynamic pressure field generated along the lubricated region. For a complete description of this problem, it is necessary to define the boundary conditions from which the pressure field develops. In this case, a pressure boundary condition ($p^*(\mathbf{x}, t)$ in the $\Gamma_p$) is applied as a Dirichlet condition.

### 2.2. Contact Mechanics

The concepts shown here were based on literature by Wriggers [36].

Contact mechanics can involve small or large deformations. Two conditions can occur when there is contact between bodies: the non-penetration and the short penetration (or simply penetration) condition. These conditions form the constraint equations of the contact problem. Thus, in the contact area of the bodies, the constraint equations for the normal and tangential contact types are formulated.

It is considered a boundary value problem for frictionless contact between a deformable surface and a rigid surface. The consideration of frictionless contact is justified by the fact that it follows the same line of research scientists from different fields, such as Hardy and Hardy [37], Holm [38], Ernst and Merchant [39], Tomlinson [40], and Bowden et al. [41]

who, through examination of the frictional properties of surfaces with different degrees of contamination, showed that the frictional force comes from the micro-scale deformation energy of the sliding contact surfaces in a dissipative process. Since there is a noticeable difference between the actual and apparent contact areas, only the actual area determines the magnitude of the frictional force. The contact conditions are:

$$
\begin{aligned}
u_N - g &\leq & 0, \\
p_N &\leq & 0 \quad \text{em} \quad \Gamma_c, \\
(u_N - g) &= & 0,
\end{aligned}
\tag{6}
$$

where $u_N = \mathbf{u} \cdot \mathbf{n}$ is the normal component of the displacement field, and $p_N$ is the contact pressure, which is equivalent to the normal component of the tensile vector $p_N = \mathbf{t} \cdot \mathbf{n}$. The tangential components are not part of the problem since the contact is considered frictionless. Furthermore, $\Gamma_c$ is the boundary region where contact occurs between the bodies, and g is the gap (distance) between them. These conditions are related to the fact that contact occurs between the parts and the penetration phenomenon. For example, the contact condition $p_N \leq 0$ refers to the fact that contact can occur ($p_N < 0$) or not ($p_N = 0$).

### 2.3. Coupled System

There are two ways to introduce lubrication into the solid-fluid problem: one is the coupled treatment of the problem. The other is related to the fact that lubrication generally reduces the coefficient of friction $f$. Thus, some parameters can be incorporated into the constitutive friction relations, best discussed in the bibliography by Wriggers [36].

When treating lubrication as a coupled problem, the relevant equations for the fluid film are given. These are derived from the classical Navier–Stokes equations, introducing several simplifications inherent to the problem, such as considering that the flow at the interface is laminar and incompressible. Additionally, if the nonlinear convective term in the Navier–Stokes equations can be neglected, as well as the inertia terms, the general equations reduce to the Reynolds equation:

$$
\frac{\partial}{\partial x}\left( \frac{h_s^3}{\mu_f} \frac{\partial p_N}{\partial x} \right) + \frac{\partial}{\partial y}\left( \frac{h_s^3}{\mu_f} \frac{\partial p_N}{\partial y} \right) = 6 \mathrm{v}_x \left( \frac{\partial p_N}{\partial x} \right)
\tag{7}
$$

This equation is valid for stationary processes when a constant relative slip velocity $\mathrm{v}_x$ is employed and for $\delta > 0$, or $g > 0$ (see Figure 1). The variable $h_s$ characterizes the height of the gap, which may depend on the deformations of the solids. Hence, a coupled nonlinear problem with the coupling terms being the contact pressure $p_N$ and the deformation-dependent height $h_s$ is constituted.

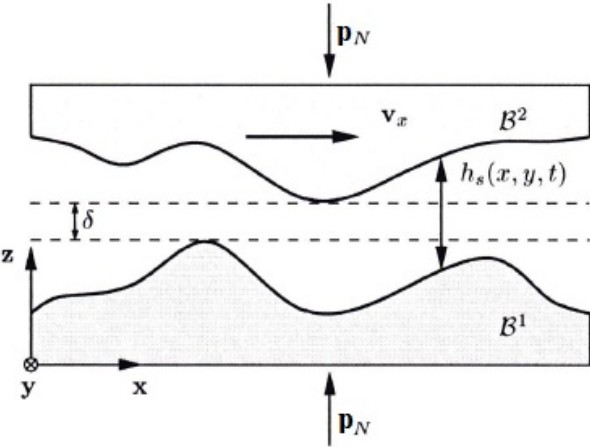

**Figure 1.** Lubrication interface definition.

According to Zienkiewicz and Taylor [42], coupled systems and formulations are those applicable to multiple domains and dependent variables that often describe different physical phenomena. No domain can be solved separately from the others, just as it can explicitly eliminate no set of dependent variables at the differential equation level.

Therefore, the dynamic fluid–structure interaction problem cannot solve the fluid and solid formulations separately due to the unknown forces at the interface. This coupling can be weak or strong, depending on the degree of interaction. Furthermore, this coupling occurs at the interfaces between domains through the boundary conditions imposed there. So, the displacement of the solid $(\mathbf{u}(\mathbf{x},t))$ influences and interacts with the generation of pressures in the fluid $\left(p_f(\mathbf{x},t)\right)$, according to the following equations.

$$\frac{\partial p_f}{\partial n} = -\rho_f \dot{\mathbf{v}}_{fn} = -\rho_f \dot{\mathbf{v}}_f \mathbf{n} \tag{8}$$

$$\dot{\mathbf{v}}_{fn} = \ddot{u}_n = \ddot{\mathbf{u}} \mathbf{n} \tag{9}$$

The weak form of the fluid in the coupled system will be given by

$$\int_{B_f} \delta p_f \nabla \left(\nabla p_f\right) dB + \int_{\Gamma_c} \delta p \ddot{\mathbf{u}} \mathbf{n} d\Gamma = 0 \tag{10}$$

where $B_f$ represents the fluid domain, the coupled problem is discretized into the standard form for the approximate displacement vector and the pressure in the fluid, using the nodal parameters of each field and the appropriate shape functions. Then, they are entered into the discrete equations for the solid and fluid, connected through the coupling term that appears due to the pressures.

## 3. Material and Methods

In Figure 2, a flowchart is presented in which the methodology employed in this work is described.

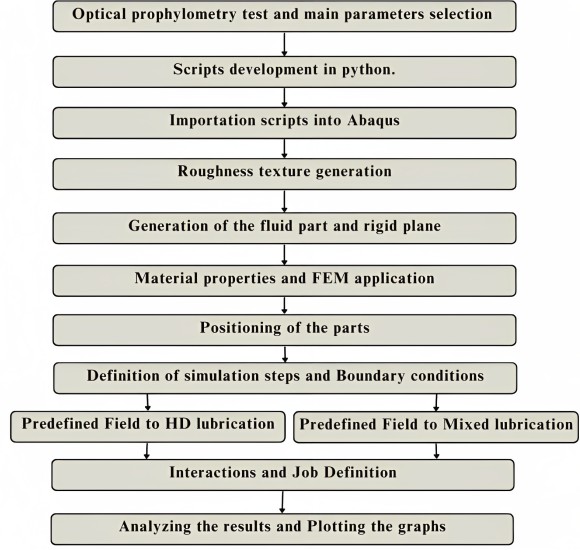

**Figure 2.** Methodology flowchart.

The computational model developed in this study represents the microscopic contact between the surfaces of components present in rolling bearings under lubrication regimes. First, the rolling surface texture data are acquired by running an optical profilometry test. The data were processed and a computational model with a realistic surface was elaborated. Then, two cases of fluid–structure iteration were simulated: sliding contact with mixed lubrication and with hydrodynamic lubrication, with varying lubricant film heights.

### 3.1. Surface Roughness

An optical profilometry test (NANOVEA PS50 Profilometer) took place to obtain data on the topography of the metallic surface of the sphere present in the bearing. This test consists of the data acquisition without physical contact with the part through the emission of a laser on the surface (see Figure 3). As seen in Khonsari's literature [14], to model a contact process, it is necessary to have adequate knowledge of the topography of the surfaces. Thus, the main roughness parameters must be adequately treated to represent the profile of the asperity heights as faithfully as possible. The requirements were met through the test performed via an optical profilometer.

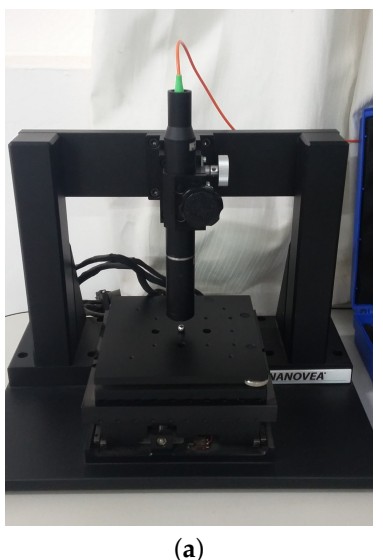
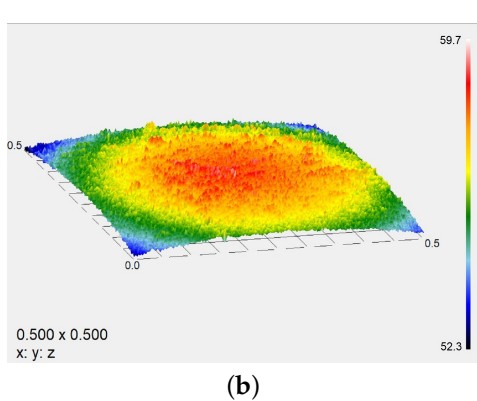

(**a**)                                                                        (**b**)

**Figure 3.** Data acquisition. (**a**) Optical profiling test of the sphere surface by 3D NANOVEA Profilometer (model PS50). (**b**) Example of roughness topography using Software NANOVEA version 3.2.7 (scan area of $0.5 \times 0.5$ mm$^2$).

The distribution of the heights (peaks and valleys) acquired through the test was the data set used in the modeling of the sphere's surface on a microscopic scale (see Figure 4). This component belongs to the bearing model KC45-58, basic size 203, shaft diameter 5/8 inches (17 mm) with standard bearing model KMB45-58-PA from the manufacturer Rexnord MB line Link-Belt Pillow Block Ball Bearings.

From the optical profilometry test, an output file of extension *.txt* is obtained through the 3D Profiler Software NANOVEA version 3.2.7. In this output file, there is the point cloud containing the x-, y- and z-coordinates of the $0.4 \times 0.4$ mm$^2$ scanned element of the ball surface. To model a surface with the flattest possible, i.e., with minimal interference from the undulation, we sought to employ the points of the scanned area of $0.4 \times 0.4$ mm$^2$. The data were treated so that the z-axis values represented the roughness parameters because the z-coordinate values in the point cloud correspond to the color scale related to the chromatic confocal measurement technique used by the profilometer.

Based on this, scripts in Python were elaborated and imported into Abaqus software that contained the x- and z-coordinates of the point cloud for each y-coordinate, creating the rough surface. In the optical profilometry test, for each value of y, the laser would travel a length in x, measuring the height of the roughness, then return to the starting point of that line and move on to the next value on the y-axis. Each imported script creates a new part of the line type. Then, these parts will be joined together to become a shell or more realistic surface.

The surface parameters calculated from the ball scan are the average roughness $R_a = 0.949$ μm and average quadratic roughness $R_q = 1.159$ μm.

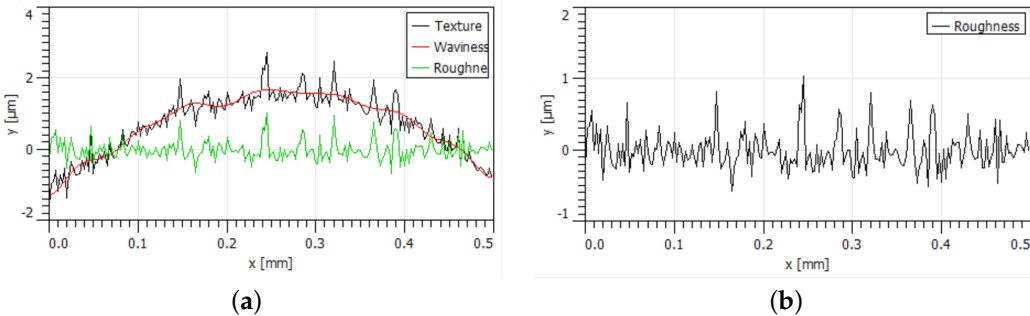

**Figure 4.** Sphere surface roughness profile. (**a**) Texture, waviness and roughness. (**b**) Roughness.

When meshing the computational model for the sample surface of $0.4 \times 0.4$ mm$^2$ to achieve analysis convergence, the number of elements was more than $10^5$, making the simulation unfeasible. So, it was decided to cut the model size and consider a final inspected area of $0.075 \times 0.075$ mm$^2$ (see Figure 5).

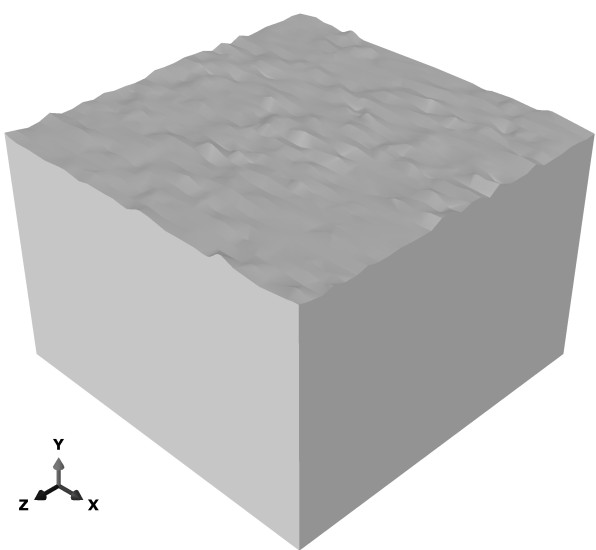

**Figure 5.** Final surface texture.

### 3.2. Computational Model

The computational model consists of three parts: a solid with a rough texture that represents the sphere surface; a perfectly smooth rigid plane, which corresponds to the surface of the outer ring of the bearing; and the last part is a fluid domain, that is, the lubricant that is present between the parts in contact.

A highlight of this model, regarding geometry, is that here a very realistic and challenging geometry was used in terms of mesh generation and adequate refinement to establish and solve the contact equations. Lorenz et al. [32] represented the rough surfaces through a sinusoidal pattern (using only the average roughness).

According to the technical data of the grease and the metal, the properties of the fluid and solid were applied, respectively, and the rigid plane was modeled as a non-deformable material without roughness.

As seen in Table 2, the values and units of the variables considered in the lubricant modeling, such as density and viscosity, comply with an isothermal regime and represent an incompressible and Newtonian viscous fluid. For the sake of simplification, since these are initial models, it is considered that the dynamic viscosity is constant. That is, it does not vary with pressure, temperature, and density throughout the simulation, presenting itself as a limitation of the models under study. Regarding the lubricant, a grease-type

with a lithium complex thickener was chosen since it is commonly used in the bearing studied. The notable work by Morales-Espejel et al. [24] performs analyses also taking into consideration this type of grease with the same kind of thickener.

Table 3 shows that the data of the isotropic linear elastic solid are complying with the mechanical properties of metal widely used in the manufacture of bearings, due to high wear resistance.

**Table 2.** Fluid properties.

| Exxon Mobil Ronex MP Grease | |
|---|---|
| Density at 24 °C (kg/mm$^3$) | $8.0 \times 10^{-7}$ |
| Dynamic viscosity at 24 °C (N.s/mm$^2$) | $2.0 \times 10^{-4}$ |
| Kinematic viscosity of basic oil at 24 °C (mm$^2$/s) | 191.67 |
| Grade NLGI | 2 |
| Thickener type | Lithium Complex |
| Visual color | Green |

MatWeb [43].

**Table 3.** Solid properties.

| Chromium Steel SAE 52100 (100Cr6) | |
|---|---|
| Modulus of elasticity (MPa) | 210,000 |
| Density (kg/mm$^3$) | $7.81 \times 10^{-6}$ |
| Poisson's ratio | 0.30 |

MatWeb [43].

One of the types of lubrication used in the model was hydrodynamic lubrication. According to Burstain's literature [13], this type occurs when the lubricant film entirely separates the surfaces so that $h >> R_a$, with film thicknesses near or more than 1 μm. Thus, to maintain a thick lubricant layer that does not allow contact between the parts, a thickness of $h = 3$ μm was considered. Another type studied was the mixed lubrication regime, considered transient between hydrodynamic and boundary. According to the same author, the thickness of the lubricant can be above 70 nm up to 1 μm, or $h \sim R_a$. In this case, it was chosen to apply $h = 1$ μm. The contact area was 5.63 μm$^2$ for both cases since the modeling is of the "all with self" type, in which all surface boundaries (solid and fluid) interact.

In both models, the fluid flow is laminar (very small Reynolds number on the order of $10^{-6}$), without effects of turbulence, Newtonian and non-compressible. These properties explain why the modeling parameters determined by the Grüneisen ratio and the Hugoniot $U_s - U_p$ curve slope are zero (shock velocity minus particle velocity). The Grüneisen ratio and Hugoniot curve slope represent the state equation for the fluid used in the software (Abaqus®). The effects of pressures, vibrations, temperatures and material properties on shock responses are easily observed by the Hugoniot curve and Grüneisen's ratio. For simplification purposes, since these are initial models, the temperature variation was not considered, i.e., it is considered constant and equal to 24 °C throughout the simulation.

Regarding contact, the analysis is purely elastic according to the Hertz approach, and no friction between asperities was considered, as the objective is to obtain the friction value as a result of the interaction between the surfaces overall [44].

The finite element mesh applied to the model presented convergence in time. The type of element used for the fluid was the linear hexahedron EC3D8R, representing an Eulerian domain and for the deformable metal surface, featuring a Lagrangian part, the type of element was the linear tetrahedron C3D4. The global and local mesh size and the total number of elements are in Table 4 and Figure 6.

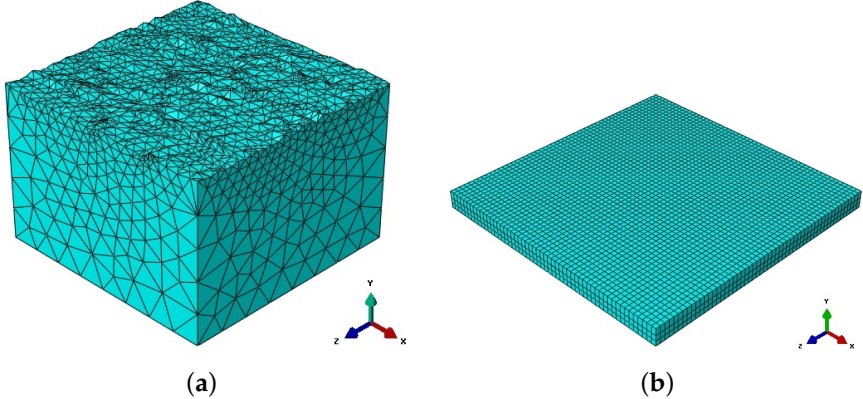

**Figure 6.** Mesh configuration. (**a**) Lagrangian mesh. (**b**) Eulerian mesh.

**Table 4.** Finite element mesh sizing.

|  | Solid | Fluid |
| --- | --- | --- |
| Global size | 0.0100 | 0.0014 |
| Local size | 0.0040 | - |
| Total number of elements | 17 793 | 8 748 |

At first, the mesh convergence in the model consisted of varying the length of the Eulerian element and thus analyzing its influence on the normal force values acting on the rough surface during normal contact (see Figure 7). In the second moment, the size of the tetrahedral elements that form the Lagrangian mesh was changed to similarly verify the behavior of the normal force acting on the rough surface (see Figure 8). The chosen size of the Eulerian element was 0.0014 mm and the reference size of the solid element was 0.004 mm. It can also be highlighted that this analysis used a higher rigor than Reichert et al. [17] related to the convergence test, pointing out the need for using smaller elements for the solid (0.01 µm × 2.5 µm) and for the fluid (0.0014 µm × 1.5 µm). The difference in the second case is alarming.

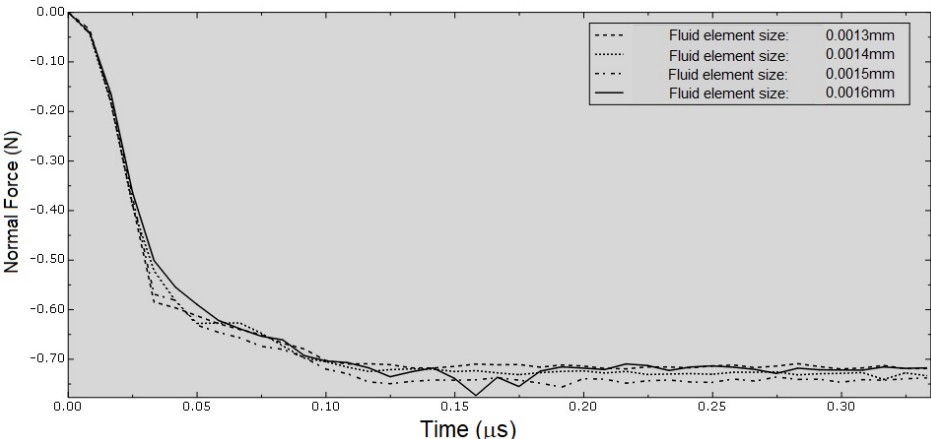

**Figure 7.** Convergence of the Eulerian mesh.

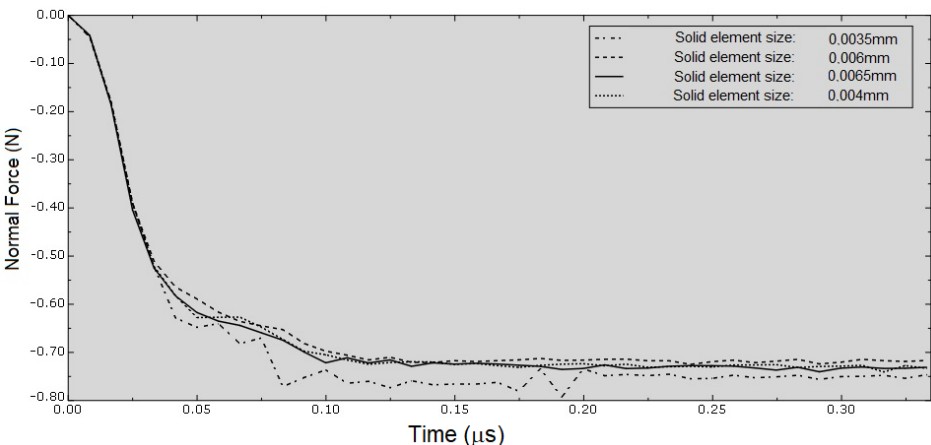

**Figure 8.** Convergence of the Lagrangian mesh.

The positioning of the parts that make up the model occurs in the Assembly mode of Abaqus®. At the bottom of the model is the solid with a rough surface, at the top is the rigid plane and the fluid domain remains confined between them. There is no penetration of fluid in the solid surface (see Figure 9).

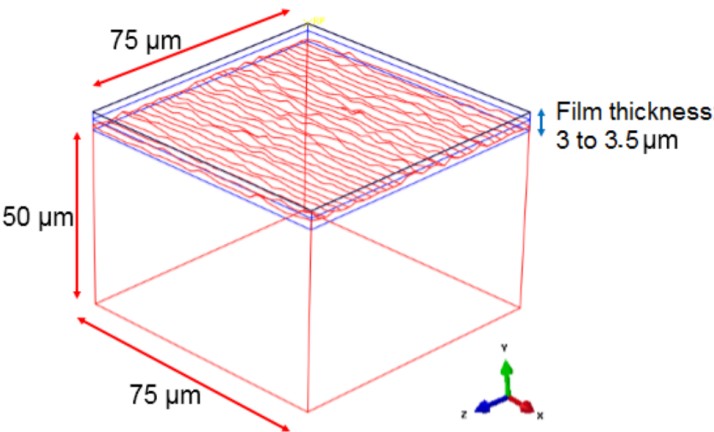

**Figure 9.** Instance positioning: rough surface (red), lubricating fluid domain (blue), and rigid plane (black).

The boundary conditions of the model seek to reproduce the real physical phenomenon. Thus, in the initial step, the speed conditions $V_{x,y,z} = 0$ mm/s are established for all parts. In the second step, this speed boundary condition for the rigid plane in the y-direction receives the value of $V_y = -3$ mm/s (displacement in the negative y-direction) to model, first, a normal contact between the solid surfaces. In the case of the hydrodynamic regime, it means maintaining the thickness of the thick lubricating film (see Figure 10).

After that, to simulate the sliding between parts, in the sliding step (displacement in the positive x-direction), the speed of the rigid plane changes to zero in the y-direction, and that of the rough surface in the x-direction changes to $V_x = 6.3$ mm/s (speed value according to Morales-Espejel et al. [24]).

Additionally, to allow fluid flow between the parts in the second and third steps, we apply a slight pressure variation to the lubricating film of $p_1 - p_2 = 110 - 109.99$ MN/mm$^2$(MPa) in the x-direction as applied by Reichert et al. [17]. The boundary conditions, defined for the rigid plane, apply at a reference point (RP) in its geometry.

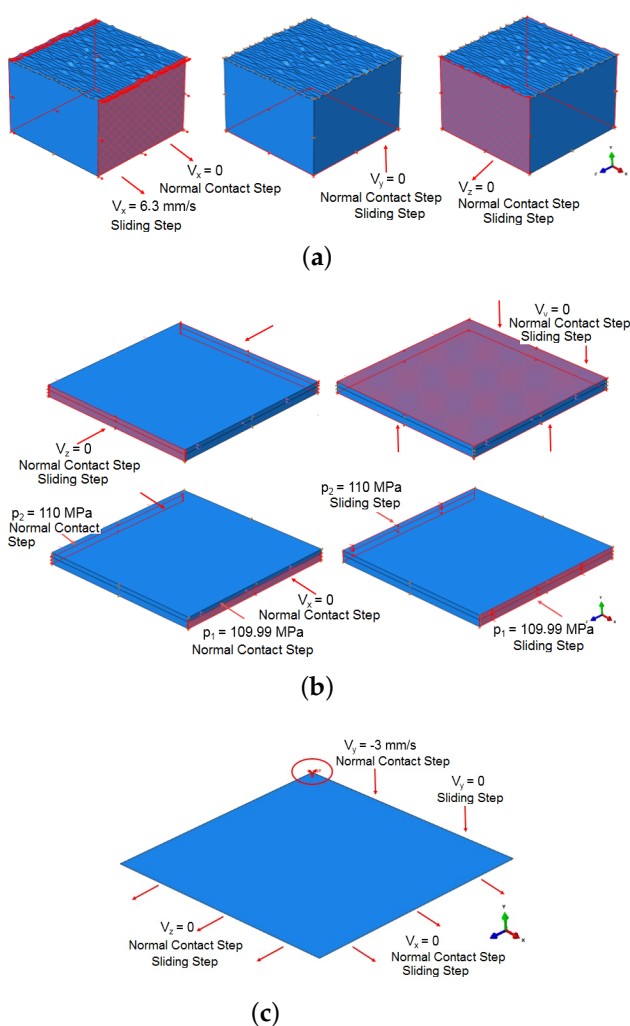

**Figure 10.** Boundary conditions of the computational model. (**a**) Solid part. (**b**) Fluid part. (**c**) Rigid part.

In summary, the operating conditions of the computational model can be seen in Tables 5 and 6.

**Table 5.** The operating conditions.

| Lubrication Regimes | Film Thickness (μm) | Reynolds Number | Temperature | Roughness $R_a/R_q$ (μm) |
|---|---|---|---|---|
| HD Lubrication | 3 | order of $10^{-6}$ | 24 °C | 1.159/0.949 |
| Mixed Lubrication | 1 | order of $10^{-6}$ | 24 °C | 1.159/0.949 |

**Table 6.** Operating conditions for both lubrication regimes.

| Steps | Solid Part | Fluid Part | Rigid Plan Part |
|---|---|---|---|
| Initial | $V_{x,y,z} = 0 \, \text{mm/s}$ | $V_{x,y,z} = 0 \, \text{mm/s}$ | $V_{x,y,z} = 0 \, \text{mm/s}$ |
| Second | $V_{x,y,z} = 0 \, \text{mm/s}$ | $V_{x,y,z} = 0 \, \text{mm/s}$ <br> $p_1 - p_2 = 110 - 109.99 \, \text{MPa}$ [2] | $V_y = -3 \, \text{mm/s}$ <br> $V_{x,z} = 0 \, \text{mm/s}$ |
| Third | $V_x = 6.3 \, \text{mm/s}$ [1] <br> $V_{y,z} = 0 \, \text{mm/s}$ | $V_{x,y,z} = 0 \, \text{mm/s}$ <br> $p_1 - p_2 = 110 - 109.99 \, \text{MPa}$ | $V_{x,y,z} = 0 \, \text{mm/s}$ |

[1] Morales-Espejel et al. [24] [2] Reichert et al. [17]

In summary, in search of expressing more clearly the conditions of operations in the modeling proposed in this paper, the following simplifying assumptions were considered:

- Incompressible and Newtonian viscous fluid.
- Fluid flow is laminar.
- Dynamic viscosity is constant.
- Pressure, temperature, and density are constant.
- Adhesion effects are not considered.
- Isotropic linear elastic solid for roughness part.
- No cavitation is modeled.
- Rigid plane was modeled as a non-deformable material without roughness.
- The effects of vibrations, temperature, and material properties on shock responses are neglected.
- Quasi-static regime with a constant sliding velocity.

To calculate the coefficient of friction and evaluate its behavior on the contact surfaces subjected to lubrication regimes, a procedure for homogenizing the results in terms of normal and tangential force was applied. The coefficient of friction was obtained from the ratio between the resisting force (tangential force) and the normal forces acting on the body.

In this work, a simple but efficient homogenization implementation procedure is presented. This procedure consists, firstly, before simulation, of generating a surface set containing all the elements in contact with the fluid and the other surface. Secondly, define the history of variables (normal force, tangential force) necessary to calculate the coefficient of friction per mesh element using the surface set created earlier. Thirdly, after the simulation, you need to access the output database where the variables are available and save them in a graphical format to obtain the values at each mesh element in time. In the fourth step, it is necessary to use specific tools to calculate the sum of the variables, obtaining the resulting forces acting on the surface. Finally, calculate the ratio between tangential and normal resultant forces to obtain the homogenized friction coefficient.

### 3.3. Handling the Fluid–Structure Problem

Another highlight of this work is the use of a fluid–structure coupling strategy to solve fluid pressure and surface stresses. In contrast, Lorenz et al. [32] analytically calculated a Hertzian pressure to distribute it over the solid surface.

The following description is based on Simulia [45].

The solid-fluid contact is solved using the general contact algorithm present in the Abaqus®. The interactions defined as the general contact of type "all with self" allow delineating the connection between all or several regions of the model with a single interaction. This definition establishes the contact between Lagrangian bodies and Eulerian materials in a coupling analysis of both.

In the case of fluid-solid contact, the Coupled Eulerian-Lagrangian Method (CEL) is used. The fluid-solid interface is assumed to be a purely no-slip contact so that no relative displacement occurs between the fluid and the solid at the boundaries. The tangential components were characterized as frictionless.

The contact modeling in terms of contact pressure is done according to what is defined concerning the normal components. Thus, the default setting was considered, as seen in Figure 11. The principle is that when the surfaces are in contact, any contact pressure can be transmitted between them. If the contact pressure is reduced to zero, the surfaces separate. The separated surfaces come into contact when the clearance between them is restricted to zero.

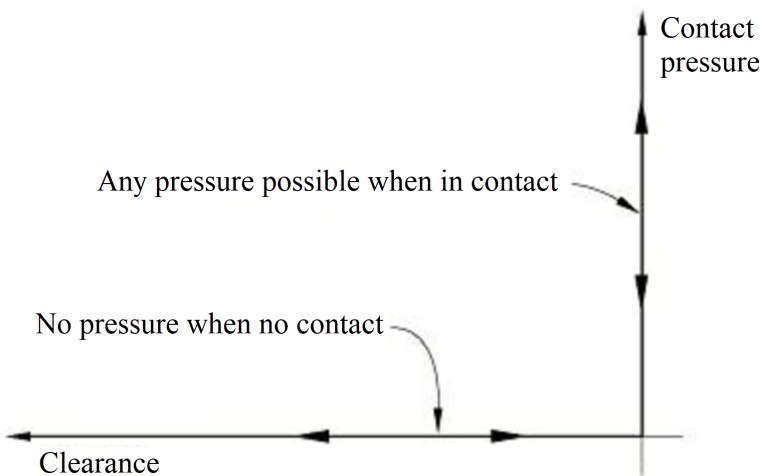

**Figure 11.** Relationship between contact pressure and clearance.

The normal contact constraints defined as "hard" are applied using the penalty method, in which the default penalty stiffness parameter is automatically maximized, subject to the stability bounds. It introduces additional stiffness behavior into the model to influence the stable time increment.

The Lagrangian–Eulerian contact formulation in Abaqus® is based on a method where the Lagrangian structure occupies empty regions within the Eulerian mesh. The general contact algorithm of Abaqus/Explicit calculates and tracks the interface between the Lagrangian-Eulerian structure defined. By employing this method, there is no need to generate a conformal mesh for the Eulerian domain, only a refinement of the elements in contact to ensure better accuracy.

Thus, at model initialization, the solid body must be positioned within the fluid mesh, and the underlying Eulerian elements must contain empty spaces. During the analysis, the Lagrangian body "pushes" the material out of the Eulerian elements through which it passes. Similarly, the Eulerian material flowing toward the Lagrangian body is prevented from entering the fluid elements filled by the solid mesh. Thus, such a formulation ensures that two materials do not occupy the same physical space.

It is allowed because implementing an Eulerian domain in Abaqus/Explicit is based on the fluid volume method. The material is tracked as it flows through the mesh, calculating its Eulerian volume fraction (EVF) within each element. While in a Lagrangian mesh, the nodes are fixed inside the material, and the elements deform as the material deforms, in an Eulerian analysis, the nodes are fixed in space, and the material flows through elements that do not warp. The amount of Eulerian material is calculated during each time increment and generally, does not correspond to an element boundary.

Newton's method is employed to solve nonlinear problems. It is based on a combination of incremental and iterative procedures. First, the solution is found by specifying the variable to be set as a time function and increment to obtain the nonlinear response. Then, the simulation is divided into many time increments, and the approximate equilibrium configuration is found at the end of each increment. Usually, many iterations are required to determine an acceptable solution for each time increment.

The time history for a simulation consists of one or more steps in which the desired analyses are defined. Each step is divided into increments so that the non-linear solution follows a path to the approximate solution. An iteration attempts to find an equilibrium solution in one increment. If the model is not in equilibrium, i.e., does not have a satisfactory solution at the end of the iteration, another iteration will be performed. The size of the time increments in this study was adjusted automatically.

## 4. Results and Discussion

The results presented are comparisons in terms of the contact pressure, fluid pressure, and friction coefficient between the two lubrication regimes considered. The first model comprises hydrodynamic lubrication, in which the film is thicker and has no proper roughness contact. The second presents the mixed regime, where the interaction may occur between some asperities, and most regions have a fluid film and no solid-solid contact.

It is possible to notice uniformly distributed contact pressures on the surface at some simulation moments. However, the contact pressure only prevails at a few peaks most of the time. It proves that due to the random dispersion of asperities on an actual rough surface, the distribution of their heights and valleys occurs in a non-homogeneous way.

The present model contains the first step toward multi-scale modeling. With the estimated average properties, one can increase its resolution. In Figure 12, it is possible to observe the dissipation of stresses in the solid to demonstrate that the proposed depth is adequate for the analysis, since the stresses at the bottom of the solid ($\sigma = 16$ MPa to $\sigma = 116$ MPa) are small compared to the maximum stress ($\sigma = 418$ MPa) close to the surface.

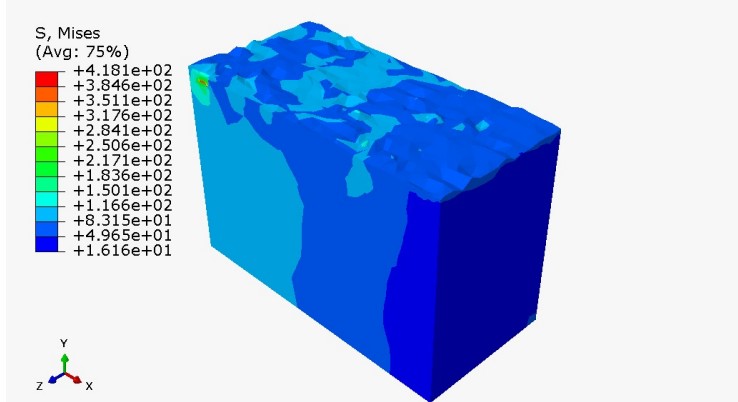

**Figure 12.** A cut view presenting the dissipation of stresses through depth.

Regarding the pressures in the fluid film, a greater variation occurs in the mixed regime, where there is greater proximity between the asperities and the rigid plane. In the hydrodynamic regime, the measurements vary between $p_f = 102$ MPa to $p_f = 169$ MPa, and in the case of mixed lubrication from $p_f = 67.5$ MPa to $p_f = 172$ MPa (see Figure 13).

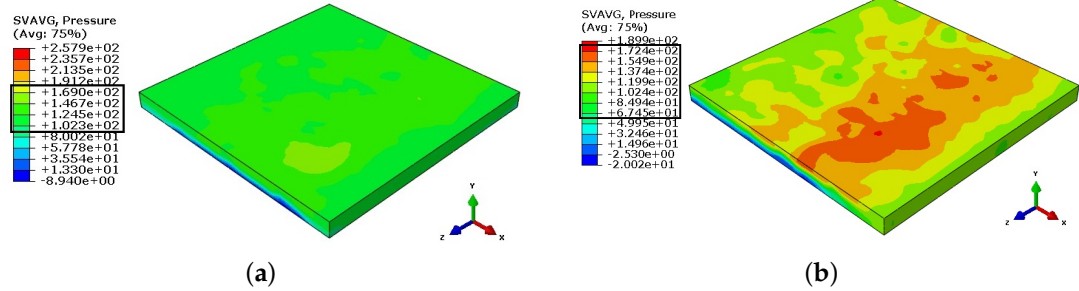

**Figure 13.** Pressure's analysis in the fluid film. (**a**) Hydrodynamic lubrication. (**b**) Mixed lubrication.

When considering this study, and the surface roughness in contact with the fluid film, it becomes appropriate to visualize the behavior of the film thickness and the pressure plots in the fluid along a central line to demonstrate more clearly the arrangement of the results found. Thus, in Figures 14 and 15, it is possible to observe said behaviors for both the mixed lubrication and hydrodynamic lubrication regimes.

Figure 14 compares the film thickness of the two lubrication regimes. It can be seen that the film thickness is almost the same, although the hydrodynamic regime started with greater fluid thickness during the working condition. At the end of the simulation, both reached the same thickness, which was expected since the rigid plane motion realized the pressure. The difference is that the film carries higher pressure under the hydrodynamic regime condition, which can be verified in Figure 15.

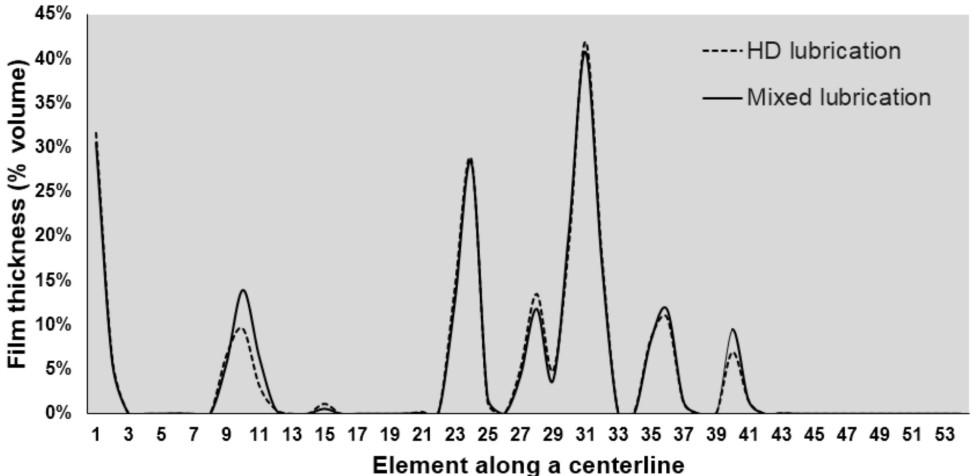

**Figure 14.** Behavior of film thickness along the central line.

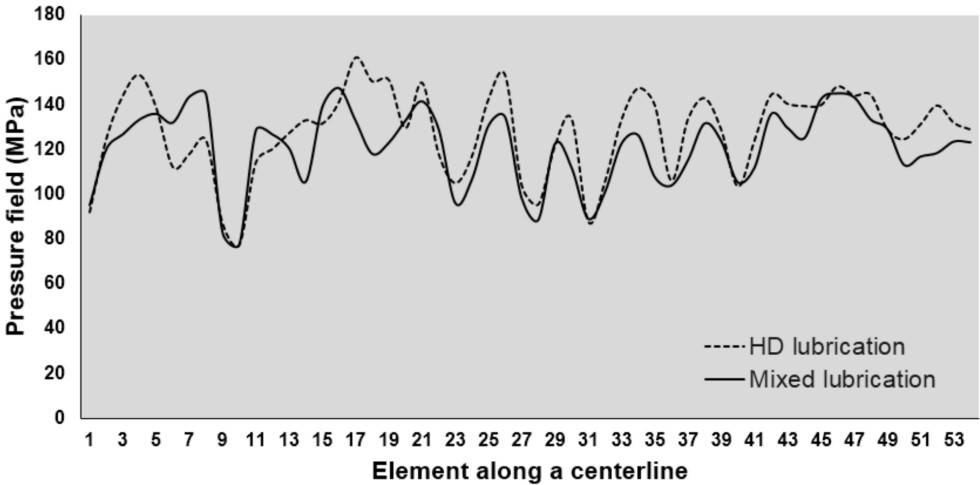

**Figure 15.** Fluid pressure profile along the central line.

In contrast, in Figure 16, the surface under the hydrodynamic regime is more protected than the surface under the mixed lubrication. Both figures being at the same scale, it can be seen that the green colors appear in a greater area on the surface, subject to a mixed film.

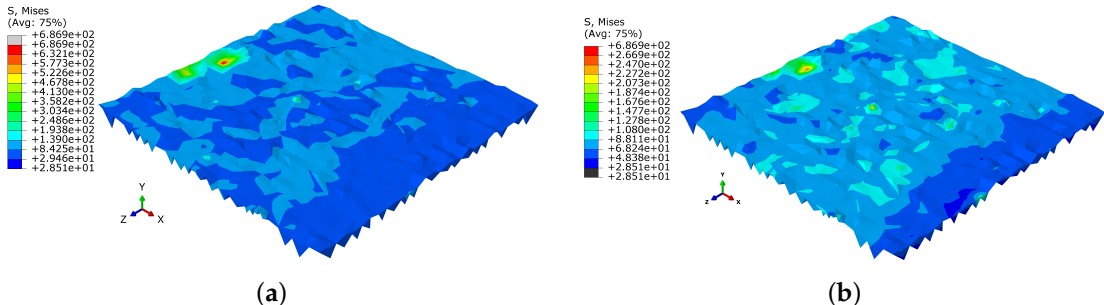

**Figure 16.** Surface stress. (**a**) Hydrodynamic lubrication. (**b**) Mixed lubrication.

Finally, a procedure to homogenize the results made it possible to obtain the friction coefficient for each time instant, according to the equation (11). It considered the ratio between the overall forces resisting movement, where $F_i^\tau$ is a tangential force at each surface element, $F_i^N$ is a normal force at each surface element and *nel* is the total number of surface elements. Thus, in Figure 17, there is an evolution of the coefficient of friction between the surfaces considering the hydrodynamic- and mixed-lubrication regimes.

$$f = \frac{\sum\limits_{i=1}^{nel} F_i^\tau}{\sum\limits_{i=1}^{nel} F_i^N} \tag{11}$$

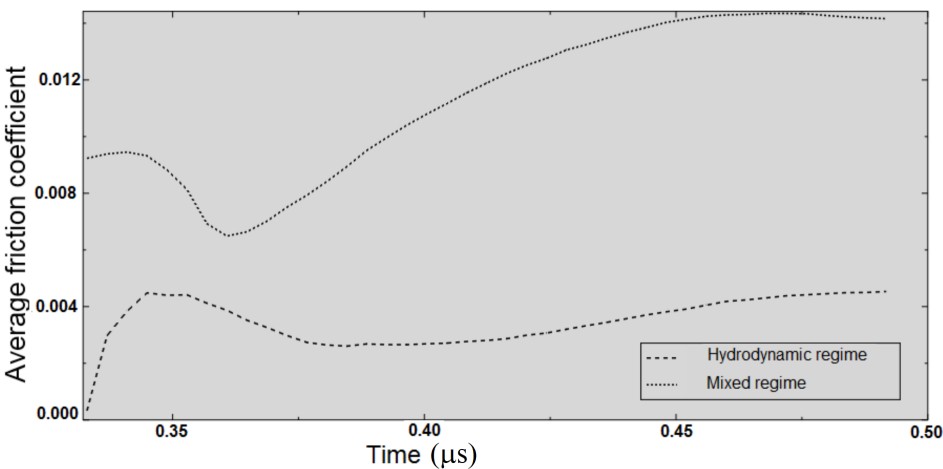

**Figure 17.** Evaluation of the friction coefficient.

It is possible to observe that in the hydrodynamic regime, the friction coefficient starts from zero at the beginning of sliding and then remains close to 0.0045. In the mixed regime, the sliding movement begins with a higher friction coefficient value and remains so, reaching close to 0.014. From qualitative analysis, this behavior concerning the two types of lubrication studied is probably in agreement with the Stribeck curve, widely used to validate studies with tribo systems; Pusterhofer and Grün [19] for example, as seen in the excellent literature review by Khonsari et al [14]. In the Stribeck curve, the friction coefficient is low, around 0.001 to 0.005, when hydrodynamic lubrication is applied, while in mixed lubrication, its value is higher, varying between 0.005 and 0.015. Therefore, it is possible to state that the values obtained in the simulation are within the range of each regime.

It is also interesting that the behavior of the friction coefficient curves obtained in this work resembles those obtained in Blau's experimental and analytical study [15,46]. For example, the curve obtained for the hydrodynamic lubrication regime is close to Blau's

curve obtained in the case of temperature increase and boundary lubrication. Additionally, the curve for the mixed lubrication regime approaches Blau's curve obtained in the case of changes in contact geometry.

The modeling using finite elements is a dynamic model. However, the working conditions are determined to reproduce a quasi-static regime with a constant sliding velocity. Thus, Table 7 shows the operational cost in terms of simulation time and the total number of increments. The steps simulation times definition is according to the choice of distance and speed for the contact and sliding of the parts. The values shown in Table 7 are for knowledge purposes, mainly regarding the time the machine takes to simulate the numerical–computational modeling developed.

**Table 7.** Simulation time.

|  | Step1 ($\mu$s) | Step2 ($\mu$s) | Total ($\mu$s) | Machine (min) | Total of Increments |
|---|---|---|---|---|---|
| HD Lubrication | 0.333 | 0.159 | 0.492 | 531 | 385,167 |
| Mixed Lubrication | 0.333 | 0.159 | 0.492 | 687 | 460,429 |

## 5. Conclusions

According to the roughness test performed, the model developed, and the associated simulations, it was possible to evaluate that:

- This paper is a mainly theoretical research paper of high accuracy, dealing with microscopical surface models to investigate the influence of the hydrodynamic and mixed lubrication on the contact of the involved surfaces in the case of a ball bearing.
- The research is mainly theoretical and proposes a new rigid surface model to investigate the friction phenomena in hydrodynamic and mixed contact modes that help understand the phenomenon. Others can use that to study its applications.
- As seen, an optical profilometry test of the surfaces of the components of a bearing, such as a sphere and the inner and outer raceways, made it possible to develop a more realistic microscopic-scale 3D model of the rough surface of the solid part of the solid–fluid contact model.
- The modeling consisted of developing and simulating hydrodynamic and mixed lubrication regimes between the parts in contact and found satisfactory results, especially when reporting friction coefficients as predicted in the literature, estimated through the contact forces analyzed in the simulations.
- The problem addressed here has complex mathematical and numerical formulations. In computational terms, when using the finite element method to solve the PDEs that govern the problem, there is the challenge of adopting a good mesh that allows achieving approximate results in less simulation time.
- The software (Abaqus®) is robust and covers several applications and modeling configurations that can further contribute to this study.

In this study, the 3D microscopic scale computational model implemented in the finite element program can analyze stresses, pressures, and friction between the lubricated bearing parts in contact using the fluid–structure interaction (FSI) method. However, there are gaps in the computational model, both associated with the geometry and the boundary conditions and the formulation of the modeling steps. Therefore, further study is needed to evaluate the complex fluid–structure interaction problem using this software.

This model is the basis for implementations in future work, in which elastoplastic properties of the solid material, thermal analysis of the contact, and variation of the viscosity with pressure and temperature must be considered. Noise vibration and harshness can also be an application area for an improved model.

**Author Contributions:** Conceptualization, F.B. and S.V.; formal analysis, M.A.; investigation, M.A.; methodology, M.A., F.B. and S.V.; project administration, F.B. and S.V.; resources, M.A.; software, F.B.; validation, M.A. and S.V.; visualization, M.A. and F.B.; writing—original draft, M.A.; writing—review and editing, F.B. and S.V. All authors have read and agreed to the published version of the manuscript.

**Funding:** This study was financed in part by the Coordenação de Aperfeiçoamento de Pessoal de Nível Superior—Brasil (CAPES)—Finance Code 001; the support from the Juiz de Fora Federal University, and Federal Institute of Southeastern Minas Gerais Juiz de Fora Campus.

**Data Availability Statement:** Not applicable

**Acknowledgments:** We would also like to thank the support from the Multidisciplinary Laboratory for Research, Development and Innovation in Theoretical and Applied Physics of the Federal Institute of Southeastern Minas Gerais Juiz de Fora Campus, Projeto CAPES Consolidacao 3 e 4-Processo 88881.708850/2022-01.

**Conflicts of Interest:** The authors declare no conflicts of interest.

## Abbreviations

The following abbreviations are used in this manuscript:

| | |
|---|---|
| EHD | Hydrodynamic lubrication |
| SRR | Slide-roll ratios |
| RMS | Root-Mean-Square |
| EHL | Elastohydrodynamic lubrication |
| CFD | Computational fluid dynamics |
| JBM | Journal bearing machine |
| FSI | Fluid–structure interaction |
| FEM | Finite element method |
| CDM | Continuous damage mechanics |
| FEA | Finite element analysis |
| RP | Reference point |
| CEL | Coupled Eulerian–Lagrangian |
| EVF | Eulerian volume fraction |

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
