# Peer review of "Fluid–Structure Interaction Analysis in Ball Bearings Subjected to Hydrodynamic and Mixed Lubrication"

_applsci, doi:10.3390/app13095660_

Round 1

Reviewer 1 Report

Comments

1. In the abstract’s last line, ‘effects of lubrication on the ‘roughness surface contact’. Will it be surface roughness contact, or is it ok?

2. In keywords: ‘lubricated’? What does it mean?

3. Line 15: Why does the citation start directly from [5] in the introduction?

4. Need to correct the citation pattern in increasing order. Start from [1], [2].

5. Line 34, which other studies? Pls, cite.

6. Line 50-62 – There are general statements from the work of previous researchers. However, something more in terms of these findings would be valuable.

7. Line 74: And Laurentis et al.[7] analyzed. No need of ‘And’. Use any other connector.

8. In the introduction section, some research gap should be written with a table. Just writing the previous work is not good.

9. Equation 5- How the boundary conditions have been taken. Pls, explain.

10. In Fig. 2. – Why ‘definition of boundary conditions’ has been repeated twice in the block?

11. Line 226- For Fig. 3. Pls also write the ‘model of the instrument with the company name’ in the sentences that explain Fig.

12. Table 1: What is the unit of kinematic viscosity?

13. Why lithium complex thickener was chosen?

14. Line 316- Vx = 6.3mm/s. Give space after number before units at all places.

15. Line 413: How homogenization was done.

15. Overall, there is also a need to explain the findings in practical, theoretical terms. Language and overall presentation need to be improved.

Thank You. 

Minor improvement is required.

Author Response

Author's Reply to the Review Report (Reviewer 1)

  1. In the abstract’s last line, ‘effects of lubrication on the ‘roughness surface contact’. Will it be surface roughness contact, or is it ok?

Author’s response: This was fixed.

Rewriting Lines 8-9: “Results demonstrate the computational model allows examining the effects of lubrication on the surface roughness contact.”

  1. In keywords: ‘lubricated’? What does it mean?

Author’s response: This was fixed.

Rewriting Lines 10-11: “Keywords: Computational modeling; Finite Element Method (FEM); Rough surface contact; Lubrication; Friction analysis.”

  1. Line 15: Why does the citation start directly from [5] in the introduction?

Authors' response: This was fixed.

  1. Need to correct the citation pattern in increasing order. Start from [1], [2].

Authors' response: This was fixed.

  1. Line 34, which other studies? Pls, cite.

Authors' response: It was described in the following paragraphs. However, to avoid doubts, citations have been included right after "other studies".

Rewriting Lines 34-62: “Other studies [13–22] focused on the causes of bearing failures through analytical, numerical, and experimental methodologies, which analyze in more detail the wear and its causes in the rolling motion, taking into account the surface roughness and the presence of lubrication of the parts in contact”. 

  1. Line 50-62 – There are general statements from the work of previous researchers. However, something more in terms of these findings would be valuable.

Authors' response: A table was created summarizing the findings of the main authors related to this work. It consists of data that were used in some way in the work, both in the adjustment of the model and for comparison and discussion of the results.

Rewriting Lines 117-119: The Table 1 summarizes the findings of the main authors related to this work. It consists of data that were used in some way in the work, both in the adjustment of the model and for comparison and discussion of the results.

Table 1. Summary of findings of the main authors related to this work.

  1. Line 74: And Laurentis et al.[7] analyzed. No need of ‘And’. Use any other connector.

Author's response: This was fixed.

Rewriting Lines 74-75: “Laurentis et al.[25 ] analyzed the connection between various grease components (thickener and base oil) applied in bearings with the sliding friction generated in lubricated contacts”.

  1. In the introduction section, some research gap should be written with a table. Just writing the previous work is not good.

Authors' response: A table was created summarizing the findings of the main authors related to this work. It consists of data that were used in some way in the work, both in the adjustment of the model and for comparison and discussion of the results.

Rewriting Lines 113: The Table 1 summarizes the findings of the main authors related to this work. It consists of data that were used in some way in the work, both in the adjustment of the model and for comparison and discussion of the results.

Table 1. Summary of findings of the main authors related to this work.

  1. Equation 5- How the boundary conditions have been taken. Pls, explain. 

Author's response: In this model, a pressure boundary condition is applied as a Dirichlet condition at inlet and outlet surfaces. To explain it better, a new paragraph was included in the text.

Rewriting Lines 168-172: The general solution of the equations consists primarily in determining the hydrodynamic pressure field generated along the lubricated region. For a complete description of this problem, it is necessary to define boundary conditions from which the pressure field develops. In this case, a pressure boundary condition ($p^{*}\left ( \mathbf{x},t \right )$ in the $\Gamma _{p}$) is applied as a Dirichlet condition.

  1. In Fig. 2. – Why ‘definition of boundary conditions’ has been repeated twice in the block?

Authors' response: This was fixed and a new flowchart was included in section 3, further describing the step-by-step methodology used in this work.

  1. Line 226- For Fig. 3. Pls also write the ‘model of the instrument with the company name’ in the sentences that explain Fig.

Author's response: This was fixed.

Rewriting Lines 239-240: “An optical profilometry test (NANOVEA PS50 Profilometer) took place to obtain data on the topography of the metallic surface of the sphere present in the bearing”.

Rewriting Lines 252-253: “From the optical profilometry test, an output file of extension \textit{.txt} is obtained through the 3D Profiler Software NANOVEA version 3.2.7”.

Fig 3: “Data acquisition. (a) Optical profiling test of the sphere surface by 3D NANOVEA Profilometer (model PS50). (b) Example of roughness topography using Software NANOVEA version 3.2.7”.

  1. Table 1: What is the unit of kinematic viscosity?

Author's response: This was fixed.

Rewriting Table 2: Kinematic viscosity of basic oil at 24â—¦C (mm2/s)

  1. Why lithium complex thickener was chosen?

Author's response: Justification has been included in the text.

Rewriting lines 290-293 :  Regarding the lubricant, a grease-type with a lithium complex thickener was chosen since it is commonly used in the bearing studied. The notable work by Morales-Espejel et al. [24] performs analyses taking into consideration also this type of grease with the same kind of thickener.

  1. Line 316- Vx = 6.3mm/s. Give space after number before units at all places. 

Author's response: This was fixed throughout the text.

  1. Line 413: How homogenization was done. 

Author's response: The procedure is simple. It consists basically of summing and division of element forces. However to clarify a new equation (11) was included and the explanation is in the text.

\begin{equation}

f  = \frac{\displaystyle\sum_{i=1}^{nel} F_i^{\tau }}{ \displaystyle\sum_{i=1}^{nel} F_i^{N}}

\label{eq:homog}

\end{equation}

Rewriting lines 468-472 : Finally, a procedure to homogenize the results made it possible to obtain the friction coefficient for each time instant, according to the equation (11). It considered the ratio between the overall forces resisting movement, where Fτi is tangential force at each surface element, FNi is normal force at each surface element and nel is the total number of surface elements.

  1. Overall, there is also a need to explain the findings in practical, theoretical terms. Language and overall presentation need to be improved. 

Author’s response:  It can be said that from our extensive literature review we could only find two references that use similar approaches to computationally model the lubricated contact taking into account the surface roughness: Reichert, S., Lorentz, B. & Albers [17] and Lorenz, S et al [32]. This scarcity would already be a reason to redo the investigation, seeking to contest or corroborate the previous results. However, when analyzing in detail the methodology of these authors, we can highlight four important points. They are now written in the manuscript.

Rewriting Lines 107-111: “The work by Reichert et al. [ 17] is actually closer to this one in terms of methodology, but we can highlight as a main difference the fact that they have dedicated mainly to mixed lubrication conditions while this paper seeks to elucidate the differences between the two regimes: hydrodynamic and mixed”.

Rewriting Lines 278-281: “A highlight for this model, regarding geometry, is that here a very realistic and challenging geometry was used in terms of mesh generation and adequate refinement to establish and solve the contact equations. Lorenz et al.[32] represented the rough surfaces through a sinusoidal pattern (using only the average roughness)”.

Rewriting Lines 329-332: “It can also be highlighted that this analysis used a higher rigor than Reichert et al. [ 17]  related to convergence test, pointing out the need of using smaller elements for the solid ($0.01 \mu m  x 2.5 \mu m$ ) and for the fluid ($0.0014 \mu m x 1.5 \mu m$ ). The difference in the second case is alarming”.

Rewriting Lines 381-383:“Another highlight of this works is the use of a fluid-structure coupling strategy to solve fluid pressure and surface stresses. In contrast,  Lorenz et al.[32] analytically calculated an Hertzian pressure to distribute it over the solid surface”.

We would like to thank you for taking the necessary time and effort to review the manuscript. We sincerely appreciate all your valuable comments and suggestions, which helped us in improving the quality of the manuscript.

Reviewer 2 Report

The authors have presented a well-structured and interested manuscript. However, at this stage, it was difficult to accept for publication. Below are some suggestions to strengthen the manuscript. The Reviewer is more than welcome to review again upon amending the manuscript.

Major

1.       Despite a highly comprehensive literature review, the Reviewer is struggling to understand the main novelty this manuscript provides.

a.       The abstract, introduction/literature review and conclusion does not mention how the methodology and results differ from that of literature.

b.       Once this is answered, then the manuscript would certainly be strengthened.

2.       Can the authors summarise the operating conditions in a table?

a.       Readers should be able to get an appreciation of what conditions were considered in this analysis.

b.       Especially, if such analysis were to evolve to more comprehensive analysis, such as exploring NVH problems.

c.       Can the authors also justify why such conditions were considered; was there a particular application the authors were focusing on?

                                                               i.      This will help with the novelty explanation.

3.       It would have been appropriate to have included film thickness and pressure plots along the centre line or a line of most interest to the authors.

a.       Surface plots alone, does not give total clarity on the generated pressured and film thickness profile, especially if asperities are being considered.

Minor

·         Can the authors explain in what context does literature review aid the methodology described in Figure 2.

o   In general Figure 2 does not appear to be described very well, further elaboration or references to the other parts the manuscript may be necessary.

·         Please include a nomenclature section with the authors abbreviation section.

Author Response

Author's Reply to the Review Report (Reviewer 2)

The authors have presented a well-structured and interested manuscript. However, at this stage, it was difficult to accept for publication. Below are some suggestions to strengthen the manuscript. The Reviewer is more than welcome to review again upon amending the manuscript.

Major

  1.       Despite a highly comprehensive literature review, the Reviewer is struggling to understand the main novelty this manuscript provides. The abstract, introduction/literature review and conclusion does not mention how the methodology and results differ from that of literature. Once this is answered, then the manuscript would certainly be strengthened.

Author’s response:  It can be said that from our extensive literature review we could only find two references that use similar approaches to computationally model the lubricated contact taking into account the surface roughness: Reichert et al. [17] and Lorenz, S et al [32]. This scarcity would already be a reason to redo the investigation, seeking to contest or corroborate the previous results. However, when analyzing in detail the methodology of these authors, we can highlight four important points. They are now written in the manuscript.

 Rewriting Lines 107-111: “The work by Reichert et al. [17] is actually closer to this one in terms of methodology, but we can highlight as a main difference the fact that they have dedicated mainly to mixed lubrication conditions while this paper seeks to elucidate the differences between the two regimes: hydrodynamic and mixed”.

Rewriting Lines 278-281: “A highlight for this model, regarding geometry, is that here a very realistic and challenging geometry was used in terms of mesh generation and adequate refinement to establish and solve the contact equations. Lorenz et al. [32]  represented the rough surfaces through a sinusoidal pattern (using only the average roughness)”.

Rewriting Lines 329-332: “It can also be highlighted that this analysis used a higher rigor than Reichert et al. \cite{reichert2016}  related to convergence test, pointing out the need of using smaller elements for the solid ($0.01 \mu$m $ \times  2.5 \mu$m ) and for the fluid ($0.0014 \mu$m $ \times  1.5 \mu$m ). The difference in the second case is alarming.”

Rewriting Lines 381-383: “Another highlight of this works is the use of a fluid-structure coupling strategy to solve fluid pressure and surface stresses. In contrast,  Lorenz et al. [32] analytically calculated an Hertzian pressure to distribute it over the solid surface”.

  1.       Can the authors summarise the operating conditions in a table? Readers should be able to get an appreciation of what conditions were considered in this analysis. Especially, if such analysis were to evolve to more comprehensive analysis, such as exploring NVH problems. Can the authors also justify why such conditions were considered; was there a particular application the authors were focusing on? This will help with the novelty explanation.

Author’s response: Operating conditions have been summarized in tables 5 and 6 and also a new paragraph has been added. There is no way of NVH problem to be analysed in this works once no modal analysis is incorporated in the numerical simulation. Unfortunately this work did not consider any frequency analysis vibration under the fluid behaviour, but this can be considered in future developments. A paragraph on possible future work has been included at the end of the conclusion section.

Rewriting Lines 353-365:“In summary, in search of expressing more clearly the conditions of operations in the modeling proposed in this paper, the following simplifying assumptions were considered:

-Incompressible and Newtonian viscous fluid.

- Fluid flow is laminar.

- Dynamic viscosity is constant.

 - Pressure, temperature, and density are constant.

 - Adhesion effects are not taken into account.

  - Isotropic linear elastic solid for roughness part.

  - No cavitation is modeled.

  -  Rigid plane was modeled as a non-deformable material without roughness.

  - The effects of vibrations, temperature and material properties on shock responses are neglected.

    - Quasi-static regime with a constant sliding velocity”.

  1.       It would have been appropriate to have included film thickness and pressure plots along the centre line or a line of most interest to the authors. Surface plots alone, does not give total clarity on the generated pressured and film thickness profile, especially if asperities are being considered.

 Author’s response:  The reviewer's suggestion was accepted and figs 14 and 15 were included. In addition, we changed the contact pressure plot to the mises stress plot on the surface, since contact pressure refers to the contact with the rigid plane, which due to lubrication , practically does not occur. The von Mises stresses reveal the transfer of efforts from the fluid to the surface and help to better identify the difference in behavior in the two regimes.

Fig. 14. Behavior of film thickness along of central line.

Fig. 15. Fluid pressure profile along of central line.

Fig. 16.  Surface stress. (a) Hydrodynamic lubrication. (b) Mixed lubrication.

 Rewriting Lines 457-464: “Thus, in Figure 14  and Figure 15, it is possible to observe the said behaviors for both the mixed lubrication and hydrodynamic lubrication regimes. Figure 14 compares the film thickness of the two lubrication regimes. It can be seen that the film thickness is almost the same, although the hydrodynamic regime started with greater fluid thickness during the working condition. At the end of the simulation, both reached the same thickness, which was expected since the rigid plane motion realized the pressure. The difference is that the film carries higher pressure under the hydrodynamic regime condition, which can be verified in Figure 15”.

 Rewriting Lines 465-467: “In contrast, in Figure~\ref{fig16}, the surface under the hydrodynamic regime is more protected than the surface under the mixed lubrication. Both figures being at the same scale, it can be seen that the green colors appear in a greater area on the surface, subject to mixed film”.

Minor

  1. Can the authors explain in what context does literature review aid the methodology described in Figure 2.  In general Figure 2 does not appear to be described very well, further elaboration or references to the other parts the manuscript may be necessary.

Authors' response: This was fixed and a new flowchart was included in section 3, further describing the step-by-step methodology used in this work.

2.  Please include a nomenclature section with the authors abbreviation section.

Authors' response: a nomenclature section with the authors abbreviation section was included.

"Abbreviations. The following abbreviations are used in this manuscript in line 547:

EHD  Hydrodynamic lubrication 

SRR  Slide-roll ratios

RMS  Root-Mean-Square

EHL  Elastohydrodynamic lubrication

CFD  Computational fluid dynamics

JBM  Journal bearing machine

FSI  Fluid-structure interaction

FEM  Finite element method

CDM  Continuous damage mechanics

FEA  Finite element analysis

RP  Reference point

CEL  Coupled Eulerian-Lagrangean

EVF  Eulerian volume fraction"

We would like to thank you for taking the necessary time and effort to review the manuscript. We sincerely appreciate all your valuable comments and suggestions, which helped us in improving the quality of the manuscript.

Reviewer 3 Report

Dear Colleagues, nice congratulations for this very high level mathematical modeling, computer aided analysis and presentation.

 However I have made some observations regarding only the cleearlyness of the presentation. See the attached file.

I am not an English teacher, but I qualify the presentation as one of high level accuracy, and style.

 As a consequence, maybe some of my remarks are not valid. 

Author Response

Author's Reply to the Review Report (Reviewer 3)

Suggestions 

  • Lines 8-9: You affirm: “Results demonstrate the computational model allows examining the  effects of lubrication on the roughness surface contact.” How to understand this?  On the contact between the rough surfaces, depending on the roughness? Or the  reduction of the roughness during the relative motion? Or simply, the effect of  lubrication on the contact between the rough surfaces? I guess this last is the idea. 

Authors' response: Yes, the reviewer is right. The idea is to examine the effect of lubrication on the contact between rough surfaces.

Rewriting Lines 8-9: “Results demonstrate the computational model allows examining the effects of lubrication on contact between rough surfaces”

  • Line 10: “Lubricated” I suggest as keyword ‘lubrication’. 

Authors' response: This was fixed.

Rewriting Lines 10-11: “Keywords: Computational modeling; Finite Element Method (FEM); Rough surface contact; Lubrication; Friction analysis.”

  • Line 13: You affirm “Bearings are machine components that provide the rotation of a cargo axle”  In my humble opinion, is false. Rotation is provided by a source of energy. Bearings  are the kinematical joints, so I suggest a reformulation like ‘Bearings are machine  components that support rotating machine elements, especially axles and wheels’ 

Authors' response: This was fixed.

Rewriting Lines 13-14: : “Bearings are machine components that support rotating machine elements, especially axles and wheels.”

  • Line 20: “Turbine designs aim to run for about 20 to 30 years” I suggest ‘Generally, turbines  runtime is expected by design for about 20 to 30 years’

Authors' response: This was fixed.

Rewriting Line 20: “Generally, turbines runtime is expected by design for about 20 to 30 years.”

  • Line 126: “Where ∇ is the gradient…” I suggest ‘Finally, ∇ is the gradient…’ do not start the  phrase with ‘where’. 

Authors' response: This was fixed.

Rewriting Lines 137-138: “Finally, ∇ is the gradient or Nabla operator, and ρ0 is the density or specific mass of the linear elastic solid.”

  • Figure 2.: In the 3rd element of the flowchart, there is an orthographical error. Correct “porphylometry” in ‘prophylometry’.

Authors' response: This was fixed and a new flowchart was included in section 3.

  • Line 218: “First, an optical profilometry test obtained the rolling surface texture data” Instead I  suggest ‘First, the rolling surface texture data is acquired by running an optical  profilometry test’

Authors' response: This was fixed.

Rewriting Lines 234: “First, the rolling surface texture data is acquired by running an optical profilometry test.”

  • Line 238: “…of the scanned surface of the 0.4 × 0.4 mm2 sphere” Instead I suggest ’ …of the 0.4  × 0.4 mm2 scanned element of the ball surface’

Authors' response: This was fixed.

Rewriting Lines 253-255: “In this output file, there is the point cloud containing the x, y and z coordinates of the 0.4 × 0.4 mm2 scanned element of the ball surface.”

  • Figure 4 b. I suggest color it with green, to correspond with the figure 4a. It will be more elegant.

Authors' response :Unfortunately, the graph shown in figure 4.b was automatically generated by the optical profiler software and we were unable to change the color of the line.

  • Line 251: “Because the finite element mesh applied to the computational model in Abaqus has more than 70 000 elements or more than 10 0000 elements, if better refined, it was  decided to decrease and consider a model with a final roughness profile area of  0.075 × 0.075^mm2 (see Figure 5).” I guess 100 000 elements, better write it as 10^5 elements. It was decidet o decrease what? The inspected area? 

Authors' response: The decrease was in the inspected area due to the high number of elements necessary for analysis convergence. The text was fixed.

Rewriting Lines 269-272: “When meshing the computational model for the sample surface of 0.4 × 0.4 mm2 to achieve analysis convergence the number of elements was more than 10^5, making the simulation unfeasible. So it was decided to cut the model size and consider a final inspected area of 0.075 × 0.075 mm^2 (see Figure 5).”

  • Line 304: At the bottom is the solid with a rough surface, the rigid plane positions at the bottom,  and between these two, the fluid part is.” A little bit confusing. Try to reformulate.

Authors' response: This was fixed.

Rewriting Lines 334-336: “At the bottom of the model is the solid with a rough surface, at the top is the rigid plane and the fluid domain remains confined between them. There is no penetration of fluid in the solid surface (see Figure 9)”.

  • Line 325: Instead “resisting movement” I suggest ’ resisting force’

Authors' response: This was fixed.

Rewriting Lines 368-369: “The coefficient of friction was obtained from the ratio between the resisting force (tangential force) and the normal forces acting on the body.”

  • Line 375 “Eulerian Eulerian analysis”- Eulerian analysis

Authors' response: This was fixed.

Rewriting Lines 418-421: “While in a Lagrangian mesh, the nodes are fixed inside the material, and the elements deform as the material deforms, in an Eulerian analysis, the nodes are fixed in space, and the material flows through elements that do not warp.”

  • Line 401: “In Figure 12 it is possible to observe the dissipation of stresses in the solid to  demonstrate that the proposed depth is adequate for the analyses studied in this  work, since they range from σ = 15MPa to σ = 97MPa, approximately.” If I inspect  carefully figure 12, I see on it approximately 10% of the surface having light grey green color, corresponding, according to the scale, to 204,5 MPa. Explain what is  the dependence of the proposed depth with the stresses? Why the interval You  specified make it adequate?

Authors' response: To explain the dissipation of stresses through the depth it was presented a new figure with a appropriate cut view. The depth is adequate for the analysis, since the the stresses at the bottom of solid are small compared to maximum stress close to the surface.

Rewriting Lines 446-449: “In Figure 12 it is possible to observe the dissipation of stresses in the solid to demonstrate that the proposed depth is adequate for the analysis, since the the stresses at the bottom of solid (σ = 16 MPa to σ = 116MPa) are small compared to maximum stress (σ = 418MPa) close to the surface.”

New figure caption: “A cut view presenting the dissipation of stresses through the depth”.

We would like to thank you for taking the necessary time and effort to review the manuscript. We sincerely appreciate all your valuable comments and suggestions, which helped us in improving the quality of the manuscript.

Round 2

Reviewer 2 Report

The Reviewer thanks the authors for adding explanations and amending the manuscript accordingly.